# A Rare Vitreoretinal Degenerative Disorder: Goldmann–Favre Syndrome Complicated with Choroidal Neovascularization in a Pediatric Patient

**DOI:** 10.3390/diagnostics15050622

**Published:** 2025-03-05

**Authors:** Klaudia Szala, Bogumiła Wójcik-Niklewska

**Affiliations:** 1Students’ Scientific Society, Department of Ophthalmology, Faculty of Medical Sciences in Katowice, Medical University of Silesia in Katowice, 40-055 Katowice, Poland; 2Department of Pediatric Ophtalmology, Faculty of Medical Sciences in Katowice, Medical University of Silesia in Katowice, 40-055 Katowice, Poland; bniklewska@sum.edu.pl; 3Kornel Gibiński University Clinical Centre, 40-514 Katowice, Poland

**Keywords:** Goldmann–Favre syndrome, autosomal recessive, vitreoretinal degenerative disorder, choroidal neovascularization

## Abstract

Goldmann–Favre syndrome (GFS) is a rare vitreoretinal degenerative disorder caused by mutations in the NR2E3 gene located on the short arm of chromosome 15. This condition, inherited in an autosomal recessive manner, was first described by Favre in two siblings, with Ricci later confirming its hereditary pattern. In GFS, rod photoreceptors are essentially replaced by S-cone photoreceptors. Enhanced S-Cone Syndrome (ESCS) and Goldmann–Favre syndrome are two distinct entities within the spectrum of retinal degenerative diseases, both caused by mutations in the NR2E3 gene. Despite sharing a common genetic basis, these conditions exhibit significantly different clinical phenotypes. ESCS is characterized by an excessive number of S-cones (blue-sensitive cones) with degeneration of rods and L-/M-cones, leading to increased sensitivity to blue light and early-onset night blindness. In contrast, GFS is considered a more severe form of ESCS, involving additional features such as retinal schisis, vitreous degeneration, and more pronounced visual impairment. GFS typically manifests in the first decade of life as night blindness (nyctalopia) and progressive visual acuity impairment. The clinical features include degenerative vitreous changes such as liquefaction, strands, and bands, along with macular and peripheral retinoschisis, posterior subcapsular cataract, atypical pigmentary dystrophy, and markedly abnormal or nondetectable electroretinograms (ERGs). Although peripheral retinoschisis is more common in GFS, central retinoschisis may also occur. Despite the consistent genetic basis, the phenotype of GFS can vary significantly among individuals. The differential diagnosis should consider diseases within the retinal degenerative spectrum, including retinitis pigmentosa, congenital retinoschisis, and secondary pigmentary retinopathy.

**Figure 1 diagnostics-15-00622-f001:**
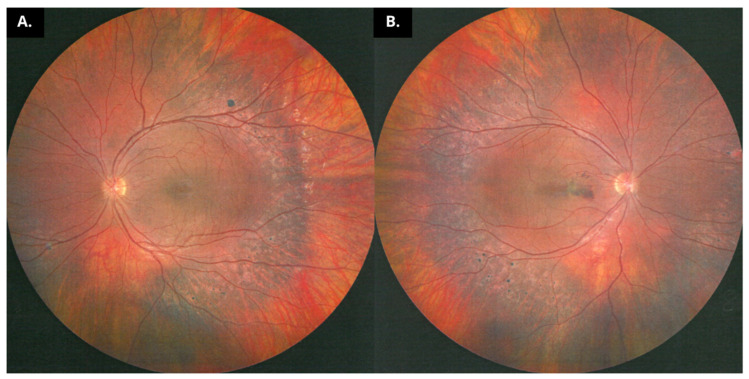
A 16-year-old patient presented to the Ophthalmology Department with a complaint of progressive vision loss observed over the past year. Since childhood, the patient has experienced difficulties with mobility and impaired night vision. Apart from ophthalmological issues, including hyperopia in both eyes, the presence of a neovascular membrane in the right eye, and astigmatism in both eyes, no other chronic diseases were diagnosed, and the family medical history remains clinically insignificant. His best corrected visual acuity (BCVA) was 0.4 in the right eye and 0.6 in the left eye. Ocular examination revealed a normal anterior segment with bilateral schisis of the macula. Typical fundus findings combined with night blindness and electroretinogram abnormalities permitted the diagnosis of Goldmann–Favre vitreoretinal degeneration [1,2]. Examination of the fundus revealed clumps of hyperplastic retinal pigment epithelium (RPE) beginning in the temporal retina, extending along the retinal vascular arcades, and reaching the optic disc in both eyes. Fundus photographs show nummular lesions with atrophic centers and hyperpigmented borders in both eyes: (**A**) left eye, (**B**) right eye. Macular edema with schisis at the fovea was observed in both eyes. The electroretinography results were abnormal, showing extinguished activity of both cones and rods [3,4]. Optical coherence tomography scans revealed hyporeflective spaces in the macular area and irregularities of the chorioretinal complex in the degenerative pigmentary areas. In cases of children with night vision disturbances, diagnostic procedures should include fluorescein angiography in addition to OCT and electrophysiological examinations. This case is particularly unique, as we describe a child with Goldmann–Favre syndrome (GFS) who has already developed choroidal neovascularization (CNV). Such an occurrence is exceptionally rare in pediatric cases of GFS, emphasizing the importance of early and comprehensive diagnostic approaches, as well as the need for vigilant monitoring in similar cases. During hospitalization, an anti-VEGF (ranibizumab) injection was administered, resulting in improved visual acuity. The patient’s clinical presentation clearly supported the diagnosis of GFS. The child was also referred to a genetic clinic for definitive confirmation of the diagnosis and is currently awaiting further testing [5,6,7]. Given the chronic and progressive nature of GFS, as well as the risk of CNV recurrence, the patient remains under regular follow-up at the Ophthalmology Clinic for ongoing monitoring and management.

**Figure 2 diagnostics-15-00622-f002:**
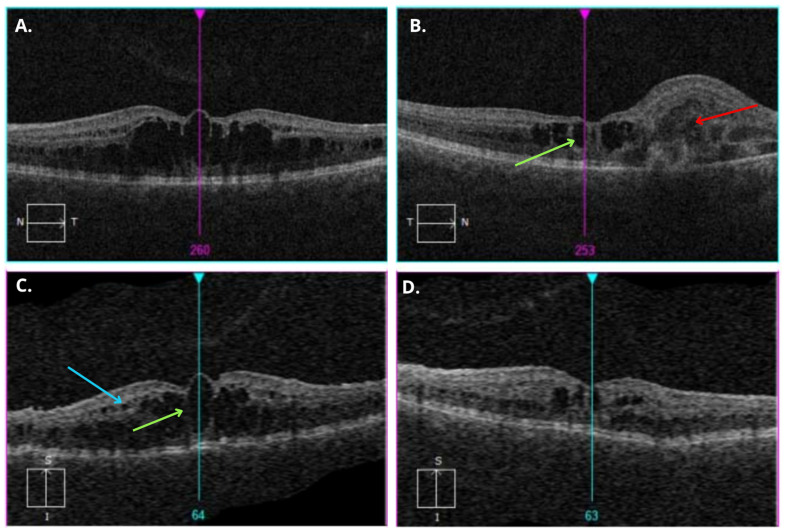
Optical coherence tomography (OCT) reveals central retinoschisis and loss of the photoreceptor layer, sudden increase in retinal thickness, and loss of the retinal laminar structure in the affected retina. Cystoid macular edema (green arrows) at the fovea is observed in both eyes. OCT images on the left (**A**,**C**), presenting patient’s left eye, showing multiple large cystic spaces and multiple macular schisis cavities in the neurosensory retina of the macula. Small cysts are visible in the inner nuclear layer (blue arrow). OCT images of patient’s right eye (**B**,**D**) (right side of the image) showing multiple cystic spaces in the neurosensory retina and chorioretinal neovascularization (CNV) in the macular area (red arrow).

**Figure 3 diagnostics-15-00622-f003:**
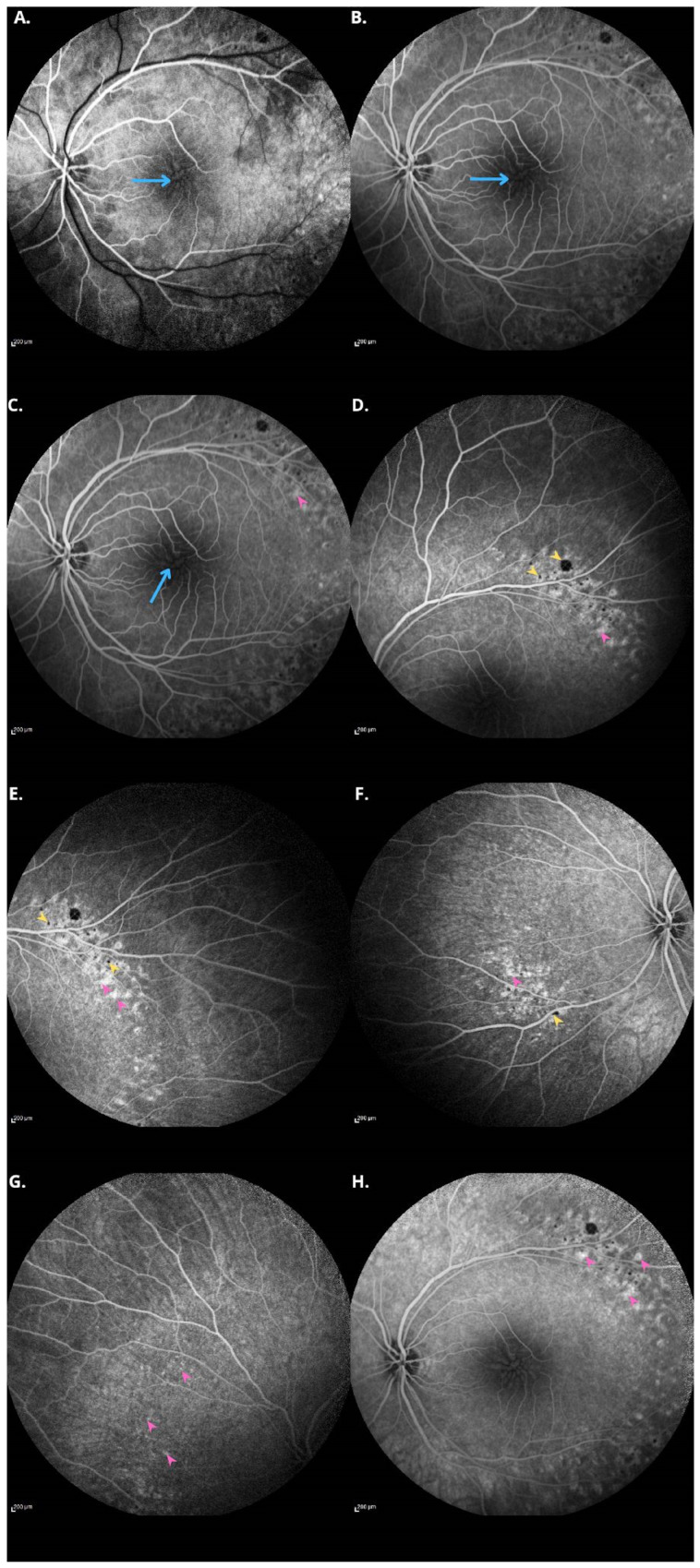
Fluorescein angiography (FA) of the left eye reveals contrast leakage on the optic disc at the 16th second. Blood flow through the main vessels is preserved. Angiographic features of non-leaking cystoid macular edema (CME) are visible, indicating the presence of retinoschisis, as highlighted by the blue arrows (**A**–**C**). In the periphery, numerous punctate foci of pigment blockage corresponding to pigmentary changes are observed, marked by yellow arrowheads, as well as areas of hyperfluorescence corresponding to window defects, indicated by pink arrowheads (**D**–**G**). In the late phase of the angiography (**H**), contrast persists in regions corresponding to the window defects.

**Figure 4 diagnostics-15-00622-f004:**
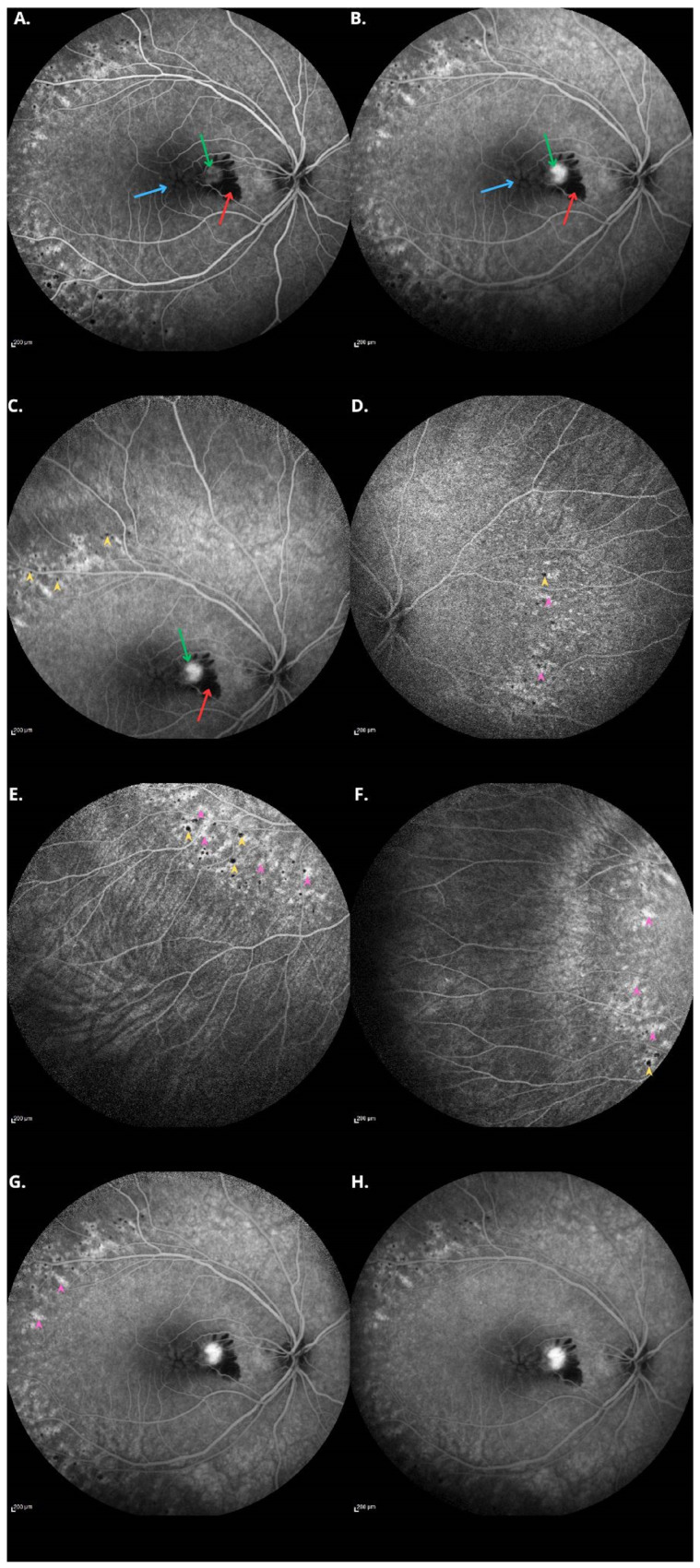
Fluorescein angiography (FA) of the right eye demonstrates preserved blood flow through the main vessels. Angiographic features of non-leaking CME are observed, indicating the presence of retinoschisis, as highlighted by the blue arrows (**A**,**B**). Additionally, in the macula, a focus of increasing hyperfluorescence over time suggests the presence of CNV, marked by green arrows. Nasal to the macula, a focus of pigment blockage corresponds to a hemorrhage, indicated by the red arrows. In the peripheral regions (**C**–**F**), numerous punctate foci of pigment blockage are visible, corresponding to pigmentary changes, as marked by yellow arrowheads. Additionally, areas of hyperfluorescence are observed, which correspond to window defects, indicated by pink arrowheads. In the late phase of the angiography (**G**,**H**), contrast persists in areas of leakage associated with CNV. This case of a child with Goldmann–Favre syndrome who developed CNV is extremely rare and highlights the importance of an individualized diagnostic and therapeutic approach in such situations. The current literature suggests that diode verteporfin photodynamic therapy (V-PDT) and intraocular anti-VEGF injections can be used either as monotherapy or in combination for the treatment of pediatric CNV. Reports indicate that anti-VEGF therapy, such as bevacizumab, has been successfully utilized following a relapse of subretinal fluid after photodynamic therapy, leading to remission of the neovascular membrane and improved visual outcomes. However, due to the rarity of CNV in pediatric patients, data on long-term efficacy and safety remain limited, highlighting the need for further research in this area [8].

## Data Availability

The original contributions presented in this study are included in the article. Further inquiries can be directed to the corresponding author.

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
