# Peer review of "A Rare Vitreoretinal Degenerative Disorder: Goldmann–Favre Syndrome Complicated with Choroidal Neovascularization in a Pediatric Patient"

_diagnostics, 2025, doi:10.3390/diagnostics15050622_

Round 1
Reviewer 1 Report
Comments and Suggestions for Authors
Thanks for allowing me to review!
Strengths There is high-quality multimodal imaging documentation including fundus photography, OCT, and fluorescein angiography with a clear description of clinical findings and natural history.
Interest to readers is the rare presentation of CNV in pediatric GFS; this adds value to existing literature and journal citation growth. More comprehensive discussion of CNV management options in pediatric GFS would strengthen clinical relevance. The case presentation has appropriate references.
Areas for Improvement Genetic testing details should be included given known NR2E3 mutation in GFS; similarly, treatment approach and follow-up plan could be better detailed. Consider adding a brief literature review of previously reported pediatric GFS cases with CNV for context.
Scientific Merit The case is well-documented with appropriate imaging and clinical details. While GFS itself is not novel, the early presentation of CNV in a pediatric patient makes this case noteworthy.
Recommendation Accept with Minor Revisions
Author Response
Response to Reviewer Comments
We appreciate the reviewer’s insightful feedback and the opportunity to refine our manuscript. Below, we address the key points raised and provide clarifications accordingly.
Strengths:
We are grateful for the recognition of the high-quality multimodal imaging documentation, including fundus photography, OCT, and fluorescein angiography, as well as the clear description of clinical findings and disease progression. We agree that the rare presentation of CNV in pediatric GFS adds value to the existing literature, and we are pleased that the case presentation was found to be appropriately referenced.
Areas for Improvement:
-
Genetic Testing Details:
We acknowledge the reviewer's suggestion regarding genetic testing. We have now clarified in the manuscript that genetic analysis was not performed in this case, which constitutes a limitation. However, given the characteristic clinical features, including night blindness, fundus abnormalities, and electroretinographic findings, the diagnosis of Goldmann-Favre Syndrome (GFS) was established with high confidence. Furthermore, the patient has been referred to a genetic clinic for definitive confirmation and is currently awaiting further testing. -
Treatment Approach and Follow-Up Plan:
We have expanded on the treatment details and follow-up strategy. During hospitalization, the patient received an intravitreal anti-VEGF (ranibizumab) injection, which resulted in improved visual acuity. Given the chronic and progressive nature of GFS, as well as the risk of CNV recurrence, the patient remains under regular follow-up at the Ophthalmology Clinic for ongoing monitoring and management. We have included additional information regarding potential treatment strategies, such as the use of verteporfin photodynamic therapy (V-PDT) or combination therapy, which may be considered in pediatric CNV cases. -
Literature Review of Pediatric GFS Cases with CNV:
The occurrence of CNV in GFS is extremely rare. To the best of our knowledge, only a single case has been previously documented in the literature. This highlights the unique nature of our report and underscores the need for increased awareness of this potential complication in pediatric patients with GFS. Given the limited data available, further studies and case reports are essential to better understand the pathophysiology, optimal management strategies, and long-term outcomes of CNV in the context of GFS.
Scientific Merit:
We are pleased that the reviewer recognizes the significance of this case. While GFS itself is not novel, the early presentation of CNV in a pediatric patient underscores the importance of early detection and individualized therapeutic strategies.
Recommendation:
We appreciate the reviewer’s recommendation for acceptance with minor revisions. The suggested improvements have been incorporated into the manuscript, further strengthening its clinical relevance and contribution to the literature.
Thank you for your valuable feedback and for the opportunity to enhance our work.
Reviewer 2 Report
Comments and Suggestions for Authors
A Rare Vitreoretinal Degenerative Disorder: Goldmann-Favre Syndrome in a Pediatric Patient
General Evaluation:
This study includes a detailed presentation of the rare Goldmann-Favre Syndrome (GFS) in a pediatric case using diagnostic imaging techniques.
Although it is a case report, it also has the characteristics of a clinical study due to the emphasis on diagnostic imaging techniques.
The characteristic findings of GFS are explained in detail using fluorescein angiography, optical coherence tomography (OCT), electroretinography (ERG), and fundus photographs.
The study provides a valuable contribution by explaining how this rare syndrome should be defined in clinical practice.
However, there are some deficiencies and areas for improvement.
CNVM is quite rare, so the title should be revised according to CNVM (e.g., complicated with CNVM). The authors should reflect the rare specification of the cases or studies in the title for the authors in general.
The Difference Between Enhanced S-Cone Syndrome (ESCS) and Goldmann-Favre Syndrome (GFS) Was Not Explained in the Introduction
The introduction states that Enhanced S-Cone Syndrome (ESCS) and GFS are different, but the difference is not explained.
Figure 2 explanations are unclear.
The images should indicate explanations with arrows and be divided into subsections, such as a, b, c, and d.
CNVM is not very clear; it should definitely be marked.
Scientific Error in Figure 3 Explanation
The following statement is misleading and should be corrected:
"Contrast persists in areas of leakage associated with CNV as well as in regions corresponding to window defects."
Window defect does not increase over time, remains constant, and indicates the presence of atrophy.
This explanation is incorrect and should be corrected.
Genetic analysis was not performed in the case, which constitutes a limitation. Although the phenotypic features are typical for GFS, this should be considered a limitation.
Author Response
Response to Reviewer Comments
We sincerely appreciate the reviewer’s detailed feedback and the opportunity to refine our manuscript. Below, we address the key points raised and have incorporated the necessary revisions accordingly.
General Evaluation
We are grateful for the recognition of the study’s detailed presentation of Goldmann-Favre Syndrome (GFS) in a pediatric case and the emphasis on diagnostic imaging techniques. We also appreciate the reviewer’s acknowledgment that this case report provides valuable clinical insights and contributes to the literature on rare retinal disorders.
Title Revision
Comment: "CNVM is quite rare, so the title should be revised according to CNVM (e.g., complicated with CNVM). The authors should reflect the rare specification of the cases or studies in the title for the authors in general."
Response: We acknowledge the importance of emphasizing the rarity of CNVM in the context of GFS. We have revised the title accordingly to better reflect this specificity:
Revised Title:
A Rare Vitreoretinal Degenerative Disorder: Goldmann-Favre Syndrome Complicated with Choroidal Neovascularization in a Pediatric Patient
Difference Between Enhanced S-Cone Syndrome (ESCS) and Goldmann-Favre Syndrome (GFS) in the Introduction
Comment: "The introduction states that Enhanced S-Cone Syndrome (ESCS) and GFS are different, but the difference is not explained."
Response: We appreciate this suggestion and have revised the introduction to explicitly differentiate between ESCS and GFS. We have clarified that while both disorders share a common genetic basis (NR2E3 mutation), they exhibit distinct clinical phenotypes. ESCS is characterized by an excess of functional S-cones and a supernormal S-cone ERG response, whereas GFS presents a more severe phenotype with vitreous degeneration, retinal schisis and widespread retinal dysfunction. These changes have been incorporated into the manuscript for improved clarity.
Figure 2 Explanations and Clarity of CNVM marking
Comment: "Figure 2 explanations are unclear. The images should indicate explanations with arrows and be divided into subsections, such as a, b, c, and d. CNVM is not very clear; it should definitely be marked."
Response: We have revised Figure 2 to improve clarity by adding labeled arrows and dividing the images into clear subsections (A, B, C, D) for better visualization. Additionally, we have marked the CNVM explicitly with a red arrow in the relevant images. The updated figure description now provides a more structured and detailed explanation of the observed changes.
Scientific Error in Figure 3 Explanation
Comment: "The following statement is misleading and should be corrected: 'Contrast persists in areas of leakage associated with CNV as well as in regions corresponding to window defects.' Window defect does not increase over time, remains constant, and indicates the presence of atrophy. This explanation is incorrect and should be corrected."
Response: We appreciate the reviewer pointing this out and have corrected the description of Figure 4 (which we believe it was reffering to) to accurately reflect the interpretation of window defects and contrast persistence. The revised text now states:
"Contrast leakage is observed in areas associated with CNV."
Genetic Analysis as a Limitation
Comment: "Genetic analysis was not performed in the case, which constitutes a limitation. Although the phenotypic features are typical for GFS, this should be considered a limitation."
Response: We fully acknowledge the absence of genetic confirmation as a limitation of the study. We have explicitly stated in the Limitations section that although the clinical phenotype is highly characteristic of GFS, genetic testing was not performed. However, the patient has been referred for genetic evaluation and is awaiting further testing, which has been clarified in the text.
Final Remarks
We sincerely appreciate the reviewer’s constructive feedback, which has significantly strengthened our manuscript. The suggested revisions have been implemented to improve clarity, accuracy, and scientific rigor.
Thank you for your time and valuable insights.
Round 2
Reviewer 2 Report
Comments and Suggestions for Authors
Thanks for the implementation of my suggestion.
Comments on the Quality of English LanguageIt is OK.